# Severe acute respiratory syndrome coronavirus (SARS-CoV-2) is not detected in the vagina: A prospective study

**Ozguc Takmaz**[1]*, **Eren Kaya**[1], **Burak Erdi**[1], **Gozde Unsal**[1], **Pari Sharifli**[2], **Nihat Bugra Agaoglu**[3], **Esra Ozbasli**[1], **Serap Gencer**[4], **Mete Gungor**[1]

1 Department of Obstetrics and Gynecology, Acibadem Mehmet Ali Aydinlar University, Istanbul, Turkey, 2 SARS-CoV-2 Laboratory Department, Turkish Directory of Health Institute, Umraniye Training and Research Hospital, Istanbul, Turkey, 3 Department of Medical Genetics, Umraniye Training and Research Hospital, Turkish Health Sciences University, Istanbul, Turkey, 4 Department of Infectious Diseases, Acibadem Mehmet Ali Aydinlar University, Istanbul, Turkey

* ozguctakmaz@hotmail.com

**Data Availability Statement:** All relevant data are within the manuscript. Minimal anonymized data set necessary to replicate our study findings was uploaded to a stable public repository. The URL of

## Abstract

### Objective

To determine whether severe acute respiratory syndrome coronavirus 2 (SARS-CoV-2) is present in the vagina of women diagnosed with coronavirus disease-19 (COVID-19) pneumonia.

### Study design

The study was conducted prospectively in a university affiliated hospital. Forty-one women of reproductive age whose nasopharyngeal PCR test were positive for SARS-CoV-2 and clinically diagnosed with pneumonia were included in the study. Vaginal swabs were obtained for SARS-CoV-2 PCR tests when the patients were admitted to the inpatient service before pneumonia treatment was initiated.

### Results

Vaginal swab samples of 38 patients were analysed with SARS-CoV-2 PCR tests. None of the vaginal swabs were positive for SARS-CoV-2.

### Conclusions

SARS-CoV-2 does not infect the vagina of women diagnosed with SARS-CoV-2 pneumonia.

## Introduction

As the name suggests, SARS-CoV-2 (severe acute respiratory syndrome coronavirus 2) is a respiratory virus [1, 2]. Knowledge on the infection of SARS-CoV-2 in systems other than the

the data set is "https://osf.io/tbk8c/?view_only=75576e82cbe648dd86979da1857bf782.

**Funding:** The author(s) received no specific funding for this work.

**Competing interests:** The authors have declared that no competing interests exist.

respiratory system is limited. In recent studies, SARS-CoV-2 has been detected in testicular tissue, urine, and feces [3–6].

Also, for the genital system, information on the infection and effect of SARS-CoV-2 is quite limited [7, 8]. To the best of our knowledge, only five studies reported analysis of the vaginal SARS-CoV-2 RT-PCR (reverse transcriptase-polymerase chain reaction) tests. Of these five studies, three reported that all vaginal swabs were negative for SARS-CoV-2 [9–11]. One study reported positive test results for the samples, and the last study reported two different test results for vaginal SARS-CoV-2 with two different test techniques [12, 13].

Infection of pathogens in the genital system can basically cause three adverse conditions; sexually transmitted infections, transmitted infection to the fetus during vaginal delivery and damage to the reproductive system in the long term. As it is a newly defined virus, it is early to determine the long-term effect of SARS-CoV-2 on the reproductive system. However, the presence of vaginal infection is especially important in terms of the delivery method decision and transmission by vaginal contact.

Additionally, another important unknown issue of the SARS-CoV-2 is the possibility of fetal transmission. Many studies reported that SARS-CoV-2 PCR tests were negative for newborns [14, 15]. However, some of them found IgM antibody positive in newborn blood samples, which can be a sign for fetal infection [8, 16]. Some studies have detected the SARS-CoV-2 in newborns, although the frequency is very low [17, 18]. Recently, a meta-analysis reported published that vertical transmission risk cannot be excluded with the existing literature. However, if vertical infection is possible, this probability is very low [19].

In our study, we aimed to determine whether SARS-CoV-2 infects the vagina of non-pregnant women of reproductive age diagnosed with COVID-19 (coronavirus disease-19) pneumonia.

## Materials and methods

This was a prospective study which was conducted in a university-affiliated hospital from June 2020 to November 2020. Written informed consents were signed by all patients. The study was approved by the Turkish Republic Ministry of Health Scientific Research Platform Ethics Committee and Acibadem University and Acibadem Healthcare Institutions Medical Research Ethics Committee (ATADEK) (ATADEK 2020-05/43). Trial was registered in clinicaltrial.gov (NCT04437940, https://clinicaltrials.gov/ct2/show/NCT04437940?cond=vaginal+covid&draw=2&rank=1)

Patients between the ages of 18–45 with positive nasopharyngeal SARS-CoV-2 RT-PCR tests and pneumonia findings were included in the study. (Fig 1) All vaginal swabs were taken before the treatment for COVID-19 was initiated. Patients who were diagnosed with viral pneumonia and negative SARS-CoV-2 RT-PCR tests, patients who had medication for SARS-CoV-2 pneumonia before a vaginal swab was taken, patients who had vaginal bleeding, patients who were in their post-menopausal period and patients who underwent hysterectomies were excluded from the study. Symptoms and the chronic medical conditions of the patients were determined by examination, performed by the attending physician (SG) of the infectious disease department. The data of the patients' chronic medical conditions, symptoms and characteristics (age, BMI) were obtained from the inpatient charts of the patients, which were filled out by the infectious disease physicians on the first day of the hospitalizations. The sample size was 56 patients, which was calculated on the basis of the positive vaginal SARS-CoV-2 test detection in 20% of COVID-19 patients with 85% power at an $\alpha$ level of 0.05. The study was planned to recruit 60 patients, factoring into consideration those who dropped out of the study. A control vaginal swab PCR test was planned for patients whose vaginal swab

SWAB ALGORITHM

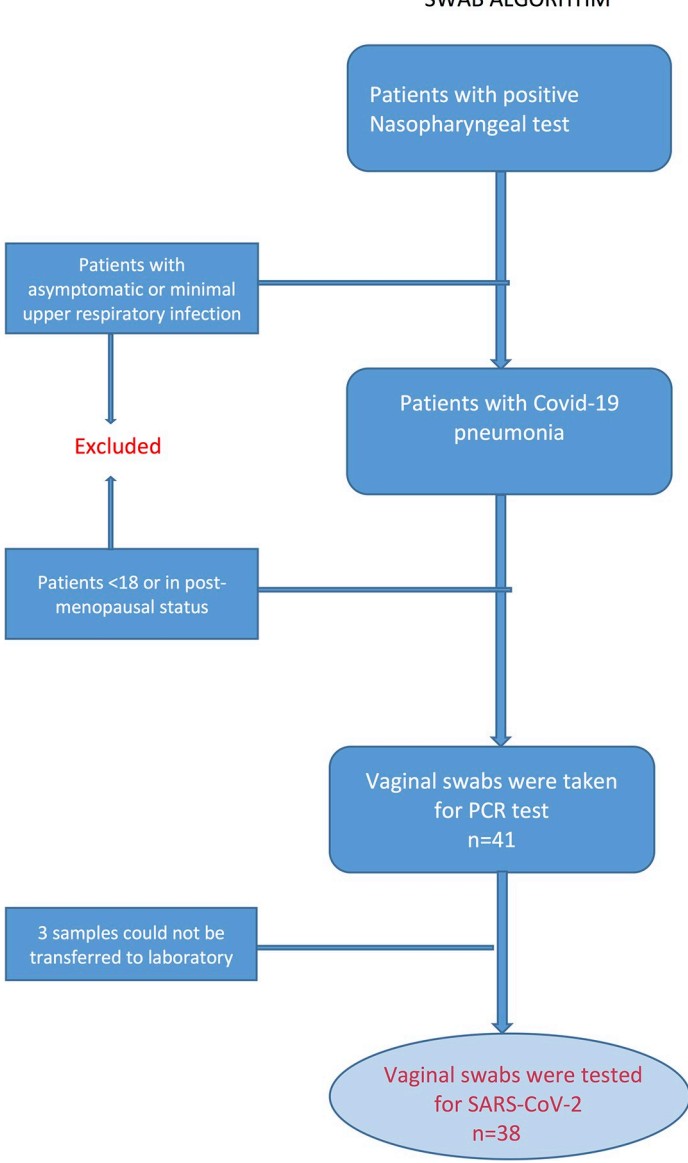

**Fig 1. Swab obtaining algorithm (41 swabs were collected but only 38 were assessed due to technical limitations).**

PCR test was positive. The first 38 patients who tested positive by nasal swab PCR had negative vaginal swabs for SARS-CoV-2, and therefore the study was terminated early.

Vaginal swabs were taken from the patients who had positive nasopharyngeal SARS-CoV-2 PCR tests and diagnosed with viral pneumonia when they were admitted to the inpatient service. (Fig 1) Swabs were inserted deep into the vagina and rotated five times. The samples were immediately transferred to the COVID-19 laboratory centers which were assigned by the Ministry of Health. Coronex SARS-CoV-2 kits (DS Bio and Nanotechnology Product Tracing and Tracking Co, Ankara, Turkey) were used for testing the samples. All tests were performed in accordance with the World Health Organization's protocols [20].

Descriptive statistics were performed by the R-3.4.3 programme (R Core Team. 2017). The assumption of normality was made by Shapiro-Wilk test. As the characteristics of the patients

distributed non-normally, the Mann-Whitney U test was used and the results were given as median with interquartile range.

## Results

Of the 41 vaginal swab samples obtained, 38 samples were tested for SARS-CoV-2. Three samples could not be sent to the laboratory due to technical conditions. Descriptive analysis of the patients' characteristics and medical conditions were performed for 38 patients. (Fig 1) Median age and BMI of the patients were 40 years (27–45) and 26.4 kg/m$^2$ (19.8–40.4), respectively (Table 1). All patients had viral pneumonia infiltration as indicated in chest computerized tomography images. The most common symptoms were fever, cough, headache, chest pain, and anosmia, respectively. Of the 38 patients, 27 had two or more symptoms. Most of the patients had no chronic medical conditions, however, hypertension was the most common determined medical condition in the study population. Diabetes, hypothyroidism, chronic ischemic heart disease, lymphoma, congenital adrenal hyperplasia, and Crohn's disease were the other reported medical disorders. Five patients had more than one chronic medical condition. Patient characteristics are presented in Table 1. None of the patients needed admission to the intensive care unit.

None of the vaginal swab samples were positive for SARS-CoV-2.

## Discussion

In our study, we investigated whether SARS-CoV-2 could be detected in the vagina of women who had COVID-19 pneumonia. None of the vaginal swab tests were positive for SARS-CoV-2. According to the results of our study, it is not expected for SARS-CoV-2 to be transmitted by sexual intercourse or delivery. However, due to limited knowledge on this subject, it is early to have a clear conclusion for genital colonisation of SARS-CoV-2.

The SARS-CoV-2 pandemic induced many scientific research projects for understanding the details of the SARS-CoV-2 pathophysiology. Although SARS-CoV-2 is a respiratory virus,

**Table 1. Characteristics and symptoms of the patients (n = 38).**

| | |
|---|---|
| Age (years) | 40 (27–45) |
| BMI* (kg/m$^2$) | 26.4 (19.8–40.4) |
| Medical Disorder | |
| Hypertension | 9 (24) |
| Diabetes | 4 (10) |
| Hypothyroidism | 2 (5) |
| Other** | 6 (16) |
| None | 22 (58) |
| COVID-19 Symptoms | |
| Fever | 24 (63) |
| Cough | 15 (39) |
| Headache | 12 (31) |
| Chest pain | 10 (26) |
| Anosmia | 9 (24) |

Data was presented as median (min-max) for age and BMI and n (%).

*BMI = Body mass index

**Other medical disorders were cardiac disease, Hodgkin lymphoma, congenital adrenal hyperplasia, Crohn's disease (Five patients had more than one medical condition).

genital effects of its infection were rarely studied. The very first study on the subject was the study of Qiu et al [9]. In their study, ten women with severe COVID-19 disease were tested for vaginal SARS-CoV-2 positivity, and none of the tests were positive. In another study, Cui et al. reported that the vaginal swabs of 35 women were negative for SARS-CoV-2 RT-PCR test [11]. In this study, 27 of the patients were positive for respiratory samples for SARS-CoV-2. Eight patients were considered to have COVID-19 pneumonia based on their clinical and laboratory findings. In both studies, the majority of the study population was in a post-menopausal status. In a post-menopausal state, oestrogen deficiency changes blood flow, pH and the thickness of the vaginal epithelium. Superficial epithelial cells are replaced by intermediate and parabasal cells [21–23]. As this change may alter infection of the SARS-CoV-2, the results of those studies may not show the infection of the SARS-CoV-2 virus in reproductive-age women. In our study, all of the patients were of reproductive age and had positive nasopharyngeal SARS-CoV-2 test results, which makes our study group more homogenous.

In another study, Aslan et al. analysed vaginal swabs for SARS-CoV-2 in pregnant women [10]. None of the samples were positive. This study gives important information for the vaginal transmission risk of SARS-CoV-2 during vaginal delivery. However, results of this study cannot exclude vaginal infection possibility of SARS-CoV-2 for women in non-pregnant reproductive age, because vaginal epithelium also changes during the pregnancy period.

Recently, Schwartz et al published their study on the subject [12]. Unlike other studies, they reported positive vaginal swab test results for SARS-CoV-2. In this study, of the total 35 patients, two (5.7%) of them were found positive for vaginal SARS-CoV-2. Of the two positive women, one was in a pre-menopausal period and the other was in a post-menopausal period. Although this result is not evidence of the vaginal transmission risk of the virus, this study showed that vagina can be infected by SARS-CoV-2.

Khoiwal et. al reported the last article on the subject [13]. They obtained 15 vaginal and 12 cervical swabs from the patients with positive nasal PCR tests, and analysed vaginal samples for SARS-CoV-2 with two different techniques. In the first technique, they performed PCR tests according to the WHO recommendations [20]. All the vaginal samples were negative for SARS-CoV-2 with this technique. In the second technique, vaginal samples were processed for transcription mediated amplification, and detection was performed by nucleic acid hybridization [13]. Of the 15 vaginal samples, three were positive resulted for SARS-CoV-2. This study revealed that different techniques can be used for detection of SARS-CoV-2 in samples which were obtained from systems other than the respiratory system.

Other important issues for SARS-CoV-2 is detecting the vertical infection risk. The absence of SARS-CoV-2 in the vagina does not mean that vertical infection cannot occur. In a meta-analysis conducted on this subject and including 29 studies, it was reported that SARS-CoV-2 can cause 3.2% vertical infection [19]. However, the evidence for the transmission of SARS-CoV-2 to the fetus during the intrauterine period is quite limited.

There are some limitations of our study. Although our study has the largest sample size in the literature, small sample size is still a limitation. Another limitation is the false negativity of RT-PCR tests. Also, the other limitation is obtaining the vaginal swabs during the initiation of clinical pneumonia, as we do not know whether vaginal infection of SARS-CoV-2 changes within the advancement of clinical disease. The main strength of our study is that it has a homogenous patient group; all women in the study were in reproductive age and also all patients had a positive nasopharyngeal SARS-CoV-2 PCR test. In addition, vaginal swabs of the patients were taken during the pneumonia period when the viral load was probably higher.

Together with our study, these five studies in the literature showed that the SARS-CoV-2 virus may not infect the vagina or may infect with a very low probability. Considering these results, it seems that vaginal transmission risk of SARS-CoV-2 by sexual intercourse or vaginal

delivery is quite low. On the other hand, in the pandemic era, performing vaginal operations such as vaginal hysterectomy, vaginal natural orifice transluminal endoscopic surgery (V-NOTES) and the other vaginal surgeries may be safe alternatives to prevent the medical staff from transmission of SARS-CoV-2 during surgical procedures.

## Conclusion

In our study, we found that SARS-CoV-2 was not detected in the vagina of reproductive-age women with COVID-19 pneumonia. When considered together with other studies, the risk of vaginal transmission of SARS-CoV-2 seems very low. However, larger studies are needed to conclude whether SARS-CoV-2 infects the vagina.

## Supporting information

**S1 Checklist. TREND statement checklist.**
(PDF)

**S1 Protocol.**
(DOCX)

**S2 Protocol.**
(DOCX)

## Acknowledgments

We thank the nurses and staff of the COVID-19 inpatient service of Acibadem Maslak Hospital. We thank Jeanette Maria Asoglu for English grammar editing of the manuscript.

## Author Contributions

**Conceptualization:** Ozguc Takmaz, Serap Gencer, Mete Gungor.

**Data curation:** Ozguc Takmaz, Eren Kaya, Burak Erdi, Gozde Unsal, Esra Ozbasli.

**Formal analysis:** Mete Gungor.

**Investigation:** Ozguc Takmaz, Pari Sharifli, Nihat Bugra Agaoglu.

**Methodology:** Ozguc Takmaz, Mete Gungor.

**Supervision:** Serap Gencer, Mete Gungor.

**Writing – original draft:** Ozguc Takmaz.

**Writing – review & editing:** Ozguc Takmaz, Mete Gungor.

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
