## [Decision Letter · Decision Letter 0]

24 Feb 2021

PONE-D-20-39173

Severe acute respiratory syndrome coronavirus (SARS-CoV-2) is not detected in the vagina

PLOS ONE

Dear Dr. Takmaz,

Thank you for submitting your manuscript to PLOS ONE. After careful consideration, we feel that it has merit but does not fully meet PLOS ONE’s publication criteria as it currently stands. Therefore, we invite you to submit a revised version of the manuscript that addresses the points raised during the review process.

The reviewers recommend that you make minor revisions to your manuscript. I therefore strongly encourage you to submit a revised manuscript that addresses the reviewer's concerns.

We look forward to receiving your revised manuscript.

Kind regards,

Martin Chtolongo Simuunza, PhD

Academic Editor

PLOS ONE

Journal Requirements:

a) Did participants provide their written or verbal informed consent to participate in this study?

Reviewers' comments:

Reviewer's Responses to Questions

**Comments to the Author**

1. Is the manuscript technically sound, and do the data support the conclusions?

Reviewer #1: Partly

Reviewer #2: Yes

Reviewer #3: Partly

2. Has the statistical analysis been performed appropriately and rigorously? 

Reviewer #1: No

Reviewer #2: N/A

Reviewer #3: Yes

3. Have the authors made all data underlying the findings in their manuscript fully available?

Reviewer #1: No

Reviewer #2: Yes

Reviewer #3: Yes

4. Is the manuscript presented in an intelligible fashion and written in standard English?

Reviewer #1: No

Reviewer #2: Yes

Reviewer #3: Yes

5. Review Comments to the Author

Reviewer #1: 1. How was the sample size of 60 arrived at?

2. English grammar needs attention. Use reported speech and check the few spelling mistakes. Do not start a sentence with a numeral or digit.

3. Inclusion criteria was 15-45 years. How come you have a 47 year-old woman?

4. Refer to the Tables and Figures in the body of your paper especially under materials and Methods and the Results section. The Tables and Figures have information that must be explained in the text.

5. Make it clear that the descriptive analysis of symtoms, medical conditions and characteristics of patients were based on the 38 women who had the vaginal swabs tested.

6. Explain the methods used to determine or assess the symptoms, medical conditions and BMI under materials and methods.

7. Make it clear if patients had more than one symptom or medical condition.

8. Would it have been better to do all the 60 women in the sample considering that it may have been a minimum risk procedure.

What was the study design? You mentioned vaginal control swabs under methods. What were these and where are they reported?

9. In Table 1, Anosmia should be 24% instead of the stated 23%.

10. Indicate the number of patients analysed in the Table 1.

11. There is no Mean and standard deviation shown in Table 1.

Reviewer #2: Introduction

Please change the term “colonization” to “infection” as it is more accurate (e.g., viruses infect, bacteria colonize).

Although the authors highlight that there is not much known about this area, the should include an additional paragraph about incidence of viral transmission from mom to baby. Recent reports of interest include below:

https://jamanetwork.com/journals/jamanetworkopen/fullarticle/2774428

https://www.nature.com/articles/s41467-020-18933-4

https://doi.org/10.1001/jama.2020.4861

https://doi.org/10.1001/jama.2020.4621

Materials and Methods

Please modify the following statement for clarity. “Patients with positive test results were planned to obtain control vaginal swabs”. Do the authors mean that patients with positive nasal swab tests were intended to undergo vaginal swabs?

Similarly, it is not clear what the authors mean in the subsequent sentence (“Owning to the fact that the test results of the first 38 patients were negative, the study was finished earlier”). Is the sentence meant to convey that the first 38 patients that tested positive by nasal swab PCR had negative vaginal swabs for SARS-CoV-2, and therefore the study was terminated early? Please clarify.

Results

In Table 1, the column numbers for medical disorder add to equal 43. Is this correct as only 41 vaginal swabs were obtained? If so, then can the authors please provide a note in the table legend that one or two patients had more than one medical condition?

Discussion

Suggest using terms like detected, transmitted, infected instead of “colonized”.

Please also consider the references suggested in the introduction as it relates to the discussion.

Figures

The authors may wish to indicate in the Figure 1 legend that 41 swabs were collected but only 38 were assessed due to technical limitations.

Reviewer #3: A prospective study was conducted in women with COVID-19 and pneumonia. They were tested for the presence of COVID in vaginal swabs. None of the swabs were positive for SARS-Cov-2.

Minor revision:

State and justify the study’s target sample size with a pre-study statistical power calculation. Typically a power calculation for a study such as this would be based on the anticipated proportion of women whose swabs test positive for COVID and the corresponding 95% confidence interval.

6. PLOS authors have the option to publish the peer review history of their article (what does this mean?). If published, this will include your full peer review and any attached files.

Reviewer #1: **Yes: **Victor Chisha Zulu

Reviewer #2: No

Reviewer #3: No

---

## [Author Response · Author response to Decision Letter 0]

5 Apr 2021

Response to Reviewers and the Academic Editor

To the Academic Editor,

We reviewed the reference list of our manuscript. Recently published a new article was discussed in the discussion section and added in the reference list (13th article in the reference list). Additionally, an erratum of a referenced article was added in the reference list (6th reference).

1- We revised the text for the journal requirements. And we are sure that our manuscript meets the PLOS ONE‘s style requirements in the revised manuscript.

2- Signed inform consents were obtained from all patients who recruited to the study. This information was added in the methods section.

Reviewer #1

1- How was the sample size of 60 arrived at?

Authors’ Response: We thank Reviewer #1 for asking this important statistical basis of our study. The sample size was 56 patients which was calculated on the basis of the positive vaginal swab Covid-19 test detection in 20% of Covid-19 patients with pneumonia, with 85% power at an α level of 0.05. The sample size was calculated in G*Power 3 (Heinrich-Heine-Universität Düsseldorf, Düsseldorf, Germany). Considering the drop out of some patients or samples, we planned to recruit 60 patients for our study. Sample size calculation was added in the methods section.

“The sample size was 56 patients which was calculated on the basis of the positive vaginal SARS-CoV-2 test detection in 20% of Covid-19 patients with 85% power at an α level of 0.05.”

2- English grammar needs attention. Use reported speech and check the few spelling mistakes. Do not start a sentence with a numeral or digit.

Authors’ Response: English grammar was checked and some spelling mistakes were corrected. Sentences which were started with numeral or digit were changed. 

3- Inclusion criteria was 15-45 years. How come you have a 47 year-old woman?

Authors’ Response: We thank Reviewer #1 for this correction, 47 was misspelled in Table 1. Table 1 was corrected.

4- Refer to the Tables and Figures in the body of your paper especially under materials and Methods and the Results section. The Tables and Figures have information that must be explained in the text.

Authors’ Response: Table and Figure were referred in the material, methods and result sections. Information which was shown in the Table 1 and Figure 1 was explained and added in the Results section.

“Descriptive analysis of patients’ characteristics and medical conditions were performed for 38 patients. (Figure1)”

“Of the thirty-eight patients, 27 had two or more symptoms.” 

“Diabetes, hypothyroidism, chronic ischemic heart disease, lymphoma, congenital adrenal hyperplasia and Crohn disease were the other reported medical disorders. Five patients had more than one chronic medical condition.”

5- Make it clear that the descriptive analysis of symptoms, medical conditions and characteristics of patients were based on the 38 women who had the vaginal swabs tested.

Authors’ Response: Thank you for this important point. We added in the results section that the descriptive data and vaginal swab of 38 patients analyzed in the study. Also, number of analyzed patients were added in the Table 1.

“Descriptive analysis of patients’ characteristics and medical conditions were performed for 38 patients. (Figure1)”

6- Explain the methods used to determine or assess the symptoms, medical conditions and BMI under materials and methods.

Authors’ Response: Symptoms, chronic medical conditions and the patient characteristics were determined by the examination performed by the infectious disease physician on the first day of their hospitalization. This information was added in the Materials and Methods section.

“Symptoms and the chronic medical conditions of the patients were determined by examination performed by the attending physician of the infectious disease department. The data of the chronic medical conditions, symptoms and the patients’ characteristics (age, BMI) were recorded from the inpatient charts of the patients, which was filled by the infectious disease physician on the first day of the hospitalization.”

7- Make it clear if patients had more than one symptom or medical condition.

Authors’ Response: Thank you for this important point. Twenty-seven patients had more than one symptoms and 5 patients had more than one chronic conditions. This information was added in the results section.

“Of the thirty-eight patients, 27 had two or more symptoms.”

“Five patients had more than one chronic medical condition.”

8- Would it have been better to do all the 60 women in the sample considering that it may have been a minimum risk procedure.

What was the study design? You mentioned vaginal control swabs under methods. What were these and where are they reported?

Authors’ Response: Before the study was started, sample size of 56 patients was calculated for reaching a 85% power, as we stated in the first answer of the letter. Considering both the negative results of vaginal samples of the first 38 patients in our study and also the negative results of other small case series reported, the authors decided to close the patient enrollment. We think the negative vaginal swab results of 38 patients would provide sufficient information on the subject. Further studies designed with a different methodology may bring new information on the subject.

 In our study design, control vaginal swabs was planned to obtain from the patients who had positive test result for vaginal SARS-CoV-2 swab test. Because of the none of the patients had positive vaginal swab test for SARS-CoV-2, we did not obtain any control vaginal swab test.

9- In Table 1, Anosmia should be 24% instead of the stated 23%.

 Authors’ Response: The percentage of patients with anosmia symptoms was changed to 24% in Table 1.

10- Indicate the number of patients analysed in the Table 1.

Authors’ Response: Number of patients analyzed (n=38) were added in the Table 1.

11- There is no Mean and standard deviation shown in Table 1

Authors’ Response: Thank you for this point. We removed ‘Mean and standard deviation’ from the Table 1.

Reviewer #2

1- Please change the term “colonization” to “infection” as it is more accurate (e.g., viruses infect, bacteria colonize).

Authors’ Response: The term ‘colonization’ was changed to ‘infection’ in the introduction section.

2- Although the authors highlight that there is not much known about this area, the should include an additional paragraph about incidence of viral transmission from mom to baby. Recent reports of interest include below…

Authors’ Response: Thank you for this important contribution to our manuscript. We added a paragraph in the introduction section for vertical transmission of SARS-CoV-2. Also, new references for the paragraph have been added to the references.

“Additionally, another important unknown aspect of the SARS-CoV-2 is the possibility of fetal transmission. Many studies reported that SARS-CoV-2 PCR tests were negative for newborns [13, 14]. However, some of them found IgM antibody positive in newborn blood sample which can be a sign for fetal infection [8, 15]. Some studies have detected the SARS-CoV-2 in newborns, although the frequency is very low [16, 17]. Recently published a meta-analysis reported that vertical transmission risk cannot be excluded with the existing literature. However, if vertical infection is possible, this probability is very low [18]. “

3- Please modify the following statement for clarity. “Patients with positive test results were planned to obtain control vaginal swabs”. Do the authors mean that patients with positive nasal swab tests were intended to undergo vaginal swabs?

Authors’ Response: We thank Reviewer #2 for allowing us to clarify our study design. The sentence was changed to “A control vaginal swab PCR test was planned for patients whose vaginal swab PCR test was positive.”

4- Similarly, it is not clear what the authors mean in the subsequent sentence (“Owning to the fact that the test results of the first 38 patients were negative, the study was finished earlier”). Is the sentence meant to convey that the first 38 patients that tested positive by nasal swab PCR had negative vaginal swabs for SARS-CoV-2, and therefore the study was terminated early? Please clarify.

Authors’ Response: We thank Reviewer #2 for his/her contribution to the explanation of our study methodology. The study was terminated early, because of the negative vaginal swab PCR test results of the first 38 patients. And the sentences was changed to “ the first 38 patients who tested positive by nasal swab PCR had negative vaginal swabs for SARS-CoV-2, and therefore the study was terminated early”

5- In Table 1, the column numbers for medical disorder add to equal 43. Is this correct as only 41 vaginal swabs were obtained? If so, then can the authors please provide a note in the table legend that one or two patients had more than one medical condition?

Authors’ Response: In the study results, the data of 38 patients was analyzed. 22 patients had no chronic medical condition, five patients had more than one medical condition. We added in the legend of Table 1 that five patients had more than one medical condition. 

6- Suggest using terms like detected, transmitted, infected instead of “colonized”.

Authors’ Response: The term ‘colonized’ was changed to ‘detected’ or ‘infected’ in the manuscript.

7- Please also consider the references suggested in the introduction as it relates to the discussion.

Authors’ Response: We added a paragraph about the vertical transmission risk of the SARS-CoV-2 in the discussion section.

“Other important issue for SARS-COV-2 is the detecting the vertical infection risk. The absence of SARS-CoV-2 in the vagina does not mean that vertical infection cannot occur. In a meta-analysis conducted on this subject and including 29 studies, it was reported that SARS-COV-2 can cause 3.2% vertical infection. However, the evidence for the transmission of SARS-COv-2 to the fetus during the intrauterine period is quite limited.”

8- The authors may wish to indicate in the Figure 1 legend that 41 swabs were collected but only 38 were assessed due to technical limitations.

Authors’ Response: The sentence ‘41 swabs were collected but only 38 were assessed due to technical limitations’ was added in the Figure 1 legend.

Reviewer #3

1- State and justify the study’s target sample size with a pre-study statistical power calculation. Typically a power calculation for a study such as this would be based on the anticipated proportion of women whose swabs test positive for COVID and the corresponding 95% confidence interval.

Authors’ Response: We thank Reviewer #3 for asking this important statistical basis of our study. The sample size was 56 patients which was calculated on the basis of the positive vaginal swab SARS-CoV-2 test detection in 20% of Covid-19 patients with 85% power at an α level of 0.05. The sample size was calculated in G*Power 3 (Heinrich-Heine-Universität Düsseldorf, Düsseldorf, Germany). Considering the drop out of some patients or samples, we planned to recruit 60 patients for our study. Sample size calculation was added in the methods section.

“The sample size was 56 patients which was calculated on the basis of the positive vaginal SARS-CoV-2 test detection in 20% of Covid-19 patients with 85% power at an α level of 0.05.”

---

## [Decision Letter · Decision Letter 1]

23 Apr 2021

PONE-D-20-39173R1

Severe acute respiratory syndrome coronavirus (SARS-CoV-2) is not detected in the vagina: a prospective study

PLOS ONE

Dear Dr. Takmaz,

Thank you for submitting your manuscript to PLOS ONE. After careful consideration, we feel that it has merit but does not fully meet PLOS ONE’s publication criteria as it currently stands. Therefore, we invite you to submit a revised version of the manuscript that addresses the points raised during the review process.

One of the reviewers has raised a few minor comments that he feels were not adequately addressed. Please and submit your revised manuscript.

We look forward to receiving your revised manuscript.

Kind regards,

Martin Chtolongo Simuunza, PhD

Academic Editor

PLOS ONE

Journal Requirements:

Reviewers' comments:

Reviewer's Responses to Questions

**Comments to the Author**

1. If the authors have adequately addressed your comments raised in a previous round of review and you feel that this manuscript is now acceptable for publication, you may indicate that here to bypass the “Comments to the Author” section, enter your conflict of interest statement in the “Confidential to Editor” section, and submit your "Accept" recommendation.

Reviewer #1: (No Response)

Reviewer #2: All comments have been addressed

Reviewer #3: All comments have been addressed

2. Is the manuscript technically sound, and do the data support the conclusions?

Reviewer #1: Partly

Reviewer #2: (No Response)

Reviewer #3: (No Response)

3. Has the statistical analysis been performed appropriately and rigorously? 

Reviewer #1: Yes

Reviewer #2: (No Response)

Reviewer #3: (No Response)

4. Have the authors made all data underlying the findings in their manuscript fully available?

Reviewer #1: Yes

Reviewer #2: (No Response)

Reviewer #3: (No Response)

5. Is the manuscript presented in an intelligible fashion and written in standard English?

Reviewer #1: No

Reviewer #2: (No Response)

Reviewer #3: (No Response)

6. Review Comments to the Author

Reviewer #1: 1. The paper needs to be edited for standard English grammar.

2. The Mean and Standard deviation appears under methods but was not used in the analysis of data. please remove this and explain or use the methods for analysis that were done.

3. Under results section correct 'swap'to 'swab' and 'chronicle'' to 'chronic'.

4. You had mentioned that 47 years was an error and removed as it was not part of the inclusion criteria. However this still appears in the under the results section. Did you actually include a 47 year old patient?

5. Regarding the positive vaginal control swab which you earlier clarified. For observational studies like the current one a control is one that does not possess the condition or disease and a case is one with the condition or disease. then in the analysis you check if the cases are associated with certain risk factors like in your case, age, BMI, medical conditions, symptoms etc. In your study, the control was taken as one with the condition or disease. please clarify.

Reviewer #2: Thank you to the authors for addressing the concerns and making the suggested changes. The manuscript is now strengthened and improved.

Reviewer #3: (No Response)

7. PLOS authors have the option to publish the peer review history of their article (what does this mean?). If published, this will include your full peer review and any attached files.

Reviewer #1: No

Reviewer #2: No

Reviewer #3: No

---

## [Author Response · Author response to Decision Letter 1]

25 May 2021

Response to Reviewers and the Academic Editor

To the Academic Editor,

First of all, all authors thank Academic Editor and Reviewers for their positive contributions to our manuscript. In the second revision, the manuscript underwent English editing by a native speaker. And all the concerns which were pointed out by the first Reviewer were addressed. 

Reviewer #1

1- The paper needs to be edited for standard English grammar.

Authors’ Response: The manuscript was edited for English grammar by a native speaker who is experienced in grammar editing. After grammar editing, the manuscript became more fluent. Thank you for this contribution to our manuscript.

2- The Mean and Standard deviation appears under methods but was not used in the analysis of data. please remove this and explain or use the methods for analysis that were done.

Authors’ Response: Thank you for this clarification of our statistical methodology. Characteristics of the patients (BMI and age) did not distribute normally in our study. The Mann-Whitney U test was used to evaluate non-normally distributed data, which were reported as median (interquartile range, minimum, maximum). Mean and standard deviation were not used for presenting the patient characteristics. We removed this statement from our statistical methodology and we added the tests that we used for analyze. 

‘’ The assumption of normality was made by Shapiro-Wilk test. As the characteristics of the patients distributed non-normally, the Mann-Whitney U test was used and the results were given as median with interquartile range.’’

3- Under results section correct 'swap'to 'swab' and 'chronicle'' to 'chronic'.

Authors’ Response: We corrected the ‘swap’ to ‘swab’ and ‘chronicle’ to ‘chronic’ in result section.

4- You had mentioned that 47 years was an error and removed as it was not part of the inclusion criteria. However this still appears in the under the results section. Did you actually include a 47 year old patient?

Authors’ Response: Authors thank Reviewer 1 for this important correction. Before the writing process of our manuscript, The Table 1 was created. In the Table 1, it was misspelled, then result section was written on the basis of the Table. In the first round of revision, we checked the ages of all patients of the study. And none of the patients was older than 45 years old. We corrected the result section for this misspelling.

5- Regarding the positive vaginal control swab which you earlier clarified. For observational studies like the current one a control is one that does not possess the condition or disease and a case is one with the condition or disease. then in the analysis you check if the cases are associated with certain risk factors like in your case, age, BMI, medical conditions, symptoms etc. In your study, the control was taken as one with the condition or disease. please clarify. 

Authors’ Response: Our study did not have any control group that did not possess SARS-CoV-2 pneumonia. In our study design, if the vaginal swab test was positive for SARS-CoV-2, a control (repeat) vaginal swab test would have been performed later in the follow up period. These patients would not be control group. However, none of the patients’ vaginal swab test was positive, therefore we did not have any repeat (control) vaginal test for any patients. If the Reviewer 1 wants to change the word ‘control’, we will change or remove it from the manuscript.

---

## [Editor Report · Decision Letter 2]

28 May 2021

Severe acute respiratory syndrome coronavirus (SARS-CoV-2) is not detected in the vagina: a prospective study

PONE-D-20-39173R2

Dear Dr. Takmaz,

We’re pleased to inform you that your manuscript has been judged scientifically suitable for publication and will be formally accepted for publication once it meets all outstanding technical requirements.

Kind regards,

Martin Chtolongo Simuunza, PhD

Academic Editor

PLOS ONE
---

## [Editor Report · Acceptance letter]

22 Sep 2021

PONE-D-20-39173R2 

Severe acute respiratory syndrome coronavirus (SARS-CoV-2) is not detected in the vagina: a prospective study 

Dear Dr. Takmaz:

I'm pleased to inform you that your manuscript has been deemed suitable for publication in PLOS ONE. Congratulations! Your manuscript is now with our production department. 

Kind regards, 

on behalf of

Dr. Martin Chtolongo Simuunza 

Academic Editor

PLOS ONE